# Effects Caused by the Ingestion of Microplastics: First Evidence in the Lambari Rosa (*Astyanax altiparanae*)

**DOI:** 10.3390/ani13213363

**Published:** 2023-10-30

**Authors:** Ana Laura Athayde Lourenço, Glaucia Peregrina Olivatto, Adijailton José de Souza, Valdemar Luiz Tornisielo

**Affiliations:** 1Ecotoxicology Laboratory, Center for Nuclear Energy in Agriculture, University of São Paulo, Piracicaba 13400-970, SP, Brazil; analaura_athayde@usp.br (A.L.A.L.); vltornis@cena.usp.br (V.L.T.); 2Department of Gynecology and Obstetrics, Ribeirão Preto Medical School, University of São Paulo, Ribeirão Preto 14049-900, SP, Brazil; 3Luiz de Queiroz College of Agriculture, University of São Paulo, Piracicaba 13418-900, SP, Brazil; adijailtonjsouza@alumni.usp.br

**Keywords:** lambari rosa, microplastics, contamination, freshwater, ecotoxicological study

## Abstract

**Simple Summary:**

Amongst aquatic animals, fish are potentially vulnerable to the ingestion—accidental or intentional—of microplastics (MPs) discarded in the environment, due to aspects such as their similarity to food and the buoyancy of these materials, as well as their attractive coloration. By analyzing toxicological parameters (mortality, malformations, and changes in weight gain) and monitoring the ingestion and excretion of MPs from two different polymers, it was noted that exposure to a diet containing these particles was responsible for causing mortality, as well as accumulation of the ingested MPs, which were not fully excreted and remained retained in the gastrointestinal tract of the lambari rosa fish. Therefore, the results of the current study should be combined with further research, covering different species, to improve understanding of the bioaccumulation of MPs and to help devise attempts at environmental mitigation, since the harmful potential of these particles is clear, especially for the aquatic organisms that ingest them.

**Abstract:**

Microplastics are a class of contaminants that pose a threat to aquatic biota, as they are easily found in aquatic ecosystems and can be ingested by a wide variety of organisms, such as fish. The lambari rosa (*Astyanax altiparanae*) is a microphage fish, which feeds on microscopic beings and particles, making it potentially susceptible to ingesting MPs discarded in the environment. In addition, this fish is of great economic and food importance, as it is used for human consumption. This study aimed to evaluate the accumulation and possible toxicological effects caused to lambari rosa (*n* = 450) by the ingestion of polyethylene (PE) and polyethylene terephthalate (PET) MPs, since the MPs of these polymers in the form of granules, fragments, and fibers are the most commonly reported in the aquatic environment. The parameters investigated here were the quantitative analysis of ingested MPs using microscopic and staining techniques, as well as the mortality rate, malformations/injuries, and impaired weight gain. At the end of the experiment, it was concluded that MPs from both polymers accumulated in the gastrointestinal tract of the lambari rosa, and that dietary exposure, especially to the PET polymer, was responsible for increasing the mortality rate in this species.

## 1. Introduction

Microplastics can be defined as particles of synthetic polymers less than 5 mm in size [1]. Due to the increasing and frequent input of these particles into the environment, and taking into account the risks that they pose to aquatic ecosystems and the potential implications for human health, they have become a target of great concern for the scientific community [2,3]. In the aquatic environment, the main classes of MPs reported are fragments, microbeads, and fibers [4,5]. The plastic polymer polyethylene (PE) is considered to be one of the main polymers produced globally, since it is present in the manufacture of plastic objects in varied sectors. It is also the most reported polymer in samples of fragment- and microbead-type MPs collected in surface waters [6]. The textile industry utilizes both natural fibers, such as cotton and wool, and synthetic fibers, such as polyester, nylon, and elastane. Synthetic fibers account for approximately 60% of the world’s fiber consumption and are widely used in various products, such as clothing and carpets [7].

The ingestion of MPs has already been reported in more than 150 species of fish, both freshwater and saltwater [8,9,10]. This ingestion can present physical risks to small animals, such as obstruction of the digestive tract, a false sense of satiety, internal injuries—such as perforated intestines and gastric rupture—intestinal alterations, and death [11,12]. In addition, organisms that ingest MPs are exposed to a wide variety of chemicals, such as organic pollutants, metals, and additives used in the manufacture of plastic [13], and the resulting biological consequences can compromise the survival, growth, reproduction, and development of these animals [14].

Due to their small size, in addition to being ingested, MPs can also bioaccumulate or be excreted under certain conditions [15]. In a study carried out by Grigorakis et al., goldfish were fed plastic microfibers obtained from the machine-washing process [16]. After approximately 6 days, a small amount of plastic microparticles had been retained in the gastrointestinal tract of these fish, and the excretion times corresponding to 50% and 90% of the particles were 10 h and 33 h, respectively. Jovanović et al. [17] also found similar results: after 45 days of exposure, the ingestion of virgin MPs did not cause imminent harm to adult sea bream (*Spaurus aurata*) and the retention of MPs in the gastrointestinal tract of the animals was low, indicating the effective elimination of MPs from the fish’s body.

On the other hand, some studies indicate that the rate of excretion depends considerably on the species of fish and the format of the ingested MP [18]. In addition, the ingestion of MPs, whether by fish or other aquatic organisms, has the potential to cause progressive accumulation in the food chain (biomagnification), of both the plastic debris and the chemical additives associated with them, even becoming a route for human exposure [19,20]. However, the current literature on this subject is inconclusive and there is little experimental evidence on the issue, especially when considering freshwater animals, so there is an evident need to evaluate further species.

Lambaris are freshwater fish that belong to the Characidae family, with the *Astyanax* genus being the most numerous. They are commonly found in rivers, lakes, and reservoirs throughout South America [21,22]. The lambari rosa (*Astyanax altiparanae*) is a species widely used as an experimental model in the fields of biology, ecotoxicology, and reproductive physiology [23,24,25], since its anatomical and behavioral characteristics allow for an interesting comparison with other similar species. In the laboratory, they have characteristics such as ease of handling, acceptance of artificial feeding, prolificacy, and a fast life cycle (reaching the adult stage in around ten weeks) [22,26]. These fish are also microphagous, i.e., they feed on microscopic beings and particles and are susceptible to ingesting MPs in the environment, whether accidentally or intentionally [27].

## 2. Materials and Methods

### 2.1. Experimental Design

The lambari rosa fry (±1.06 g) were bought from a local fishery and underwent an acclimatization process, which was carried out for 15 min in each aquarium. After 15 min, approximately 100 mL of water from the aquariums was added to the containers holding the fish. This procedure was repeated for 20 min, with 100 mL of water being added every 5 min. After the acclimatization process, the fingerlings underwent a 10-day primary quarantine period. They were then transferred to transparent glass aquariums with a capacity of 70 L and a controlled running water system. A screen cover was also put in place to prevent the fish from escaping.

The experimental design was a completely randomized design consisting of 5 treatments (Table 1) × 30 animals × 3 replications = 450 animals. The animals were equally distributed between 15 aquariums with 70 L capacity, as described above.

The concentrations of 0.1 and 1.0% simulate the concentrations of MPs found in moderately and heavily contaminated areas, respectively [28]. The experiment lasted 90 days in order to analyze the proposed parameters during all the stages of development of the lambari rosa.

### 2.2. Preparation of the Feed Used in the Lambari Rosa Diet

Virgin (unadditivated) PE polymer pellets were supplied by Braskem and post-consumer (destined for recycling and properly sanitized) PET polymer pellets were supplied by the São Carlos Chemistry Institute (IQSC)—University of São Paulo. They were then ground in a cryogenic mill to a microscopic scale of 100–350 µm. After grinding, they were mixed with tetraMin^®^ extruded feed and water to form a paste which was extruded again and used in the lambari rosa diet. The feed was supplied twice a day, with the daily total corresponding to 4% of the total mass of fish per tank (adapted from the methodology used by Massago and Da Silva [29]).

### 2.3. Analysis of the Effects of Exposing Fish to a Diet Containing MPs

The parameters observed over the 90 days of the experiment to verify the result of the fish being exposed to the diet containing MPs were mortality, malformations/injuries, and weight.

Mortality was checked daily and the dead lambaris were immediately removed from the aquariums to prevent decomposition. The mortality rate was determined according to the total number of dead individuals at the end of the experiment in relation to the total number of individuals at the beginning of the experiment and in relation to the control treatment.

After 90 days of the experiment, 5 individuals per treatment were anesthetized with benzocaine (0.1 mg/L) and examined individually under a stereomicroscope with magnification to record and observe possible morphological abnormalities and lesions in relation to the control.

Finally, at the beginning (Ti = 0) and end of the experiment (Tf = 90), 5 individuals per treatment were anesthetized with benzocaine (0.1 mg/L) and their body weight was quantified using an analytical scale [28].

### 2.4. Retention of MPs in the Gastrointestinal Tract and Elimination through Feces

To evaluate the process of ingestion and excretion of the MPs ingested by the lambari rosa, 5 fish were randomly removed from each tank at times T = 2 h, T = 24 h, and T = 48 h after feeding [17,30,31]. The experimental design was 5 fish × 5 treatments (Control (A), PE 0.1% (B), PE 1.0% (C), PET 0.1% (D), and PET 1.0% (E)) × 3 analysis times (2, 24 and 48 h) × 3 replications = 225 individuals evaluated.

These individuals were euthanized with an overdose of benzocaine (10× the ideal dose of 0.1 mg/L). The gastrointestinal tract of each individual was removed and stored at −80 °C. To analyze the excretion of plastic microparticles by the lambari rosa, feces were collected from each aquarium using a collector at the same times as mentioned above.

To remove the organic matter present in the gastrointestinal tract and feces, the biological material digestion technique was used (adapted from the methodology used by Dhimmer [32]). The technique involved the chemical dissolution of soft tissues and organic matter using potassium hydroxide PA (10% KOH). After adding 50 mL of 10% KOH, the samples were placed in an oven at 60 °C for 48 h to optimize the time taken to carry out the process. The digested samples were then filtered with a vacuum pump on a Whatmann filter of approximately 1 µm and placed in an oven at 32 °C for 24 h to dry [33]. The filters containing MPs, after being dried in the oven, were stored in Petri dishes to avoid losses and ensure quality assurance.

To identify the MPs of the residues retained after treatment with potassium hydroxide, Nile Red dye was used, which sorbs onto the plastic surface [34]. A solution of Nile Red (Sigma-Aldrich, São Paulo, Brazil) in acetone at 1 mg/mL was prepared. A total of 5 µL of this solution was used for each sample of gastrointestinal tract and feces obtained, and each sample was left in contact with the dye on an orbital shaker (100 rpm) for 30 min.

The MP particles in each sample were observed and quantified using a microscope.

The MPs from each sample were then separated and quantified.

### 2.5. Analysis of the Results

The quantitative results obtained in the toxicological and accumulative tests are presented as means ± standard deviation. Microsoft Excel 2019 was used to tabulate the data and R Core Team 2023 (https://www.R-project.org/ (accessed on 1 October 2023))) was used for the statistical analysis. Data distribution and homogeneity of variance were checked using the Shapiro–Wilk and Bartlett tests, respectively. The ANOVA was performed for the variables MPs in gastrointestinal tract (mg) and MPs in feces (mg) and Tukey’s post hoc test was applied at a significance level of 5% to test interactions between treatment groups.

## 3. Results and Discussion

### 3.1. Toxicological Parameters

#### 3.1.1. Mortality

During the first 30 days of the experiment, only one individual from Treatment C died. However, over the next 60 days, the number of dead individuals in the other treatments (B, D, and E) increased significantly, so that at the end of the 90-day experiment, Treatment E had the highest mortality rate (17.77% ± 4.7), followed by Treatment D (16.66% ± 5.6), Treatment B (13.33% ± 4.6), and Treatment C (7.77% ± 1.2). Treatment A (control) presented a mortality rate of 3.33 ± 1.0 (Figure 1).

A possible justification for this significant increase in mortality after 30 days of experimentation concerns the toxicokinetics and delayed effects of MPs on fish survival and development, with considerable lethal and sublethal effects [28].

The individuals in Treatments D and E, i.e., those that were fed PET polymer MPs at concentrations of 0.1% and 1.0%, respectively, had the highest mortality rates at the end of the 90-day experiment. This mortality can be explained by the interruption or significant reduction in feeding due to intestinal lesions or obstructions, which can lead to death. According to Ouyang et al. [35], the ingestion of MPs can alter the composition and structure of the intestinal microbiota. In addition, according to Peda et al. [36], the ingestion of MPs can cause perforation of the intestine, ulcerative lesions, gastric rupture, and even intestinal alterations, which highlights the negative impact that such physical damage can have on the feeding rates and intestinal functions of fish.

However, when considering the PE polymer, Treatment B, which contained the lowest concentration of MPs (0.1%), had a higher mortality rate than Treatment C, which contained the highest concentration (1.0%) of MPs from this same polymer. This result suggests a lower level of exposure to MPs at this concentration, probably due to a reduction in the ingestion of these particles by the fish. According to Long et al. [37], this may have occurred because in the presence of higher concentrations of MPs, they were able to select particles containing only feed, ingesting fewer feed particles containing MPs. Another pertinent hypothesis would be that, at high concentrations, MPs can form agglomerates. However, no visible agglomerates were detected in the water during the experiment.

#### 3.1.2. Weight

Considering the initial weight (T = i) and final weight (T = f) of the lambari rosa in each treatment, there were no significant variations (*p* < 0.05) in the body weight (g) of the individuals after contamination by both types of MPs and in both concentrations tested (Figure 2). This result was confirmed when the data were compared with Treatment A (control).

Contradictorily, several studies have reported a decrease in weight gain and body growth in aquatic animals after ingesting microplastics [28,38,39]. A possible justification would be insufficient feeding, leading to limited growth of these animals. Additionally, the stress of chewing and ingesting microplastics can also contribute to low energy levels and weight loss [39]. Finally, changes in eating behavior in diets containing microplastics have also been reported [40]. Nevertheless, the fish analyzed in this study did not respond to the presence of microplastics in their feed with any changes in feeding behavior.

#### 3.1.3. Malformations

At the end of the 90-day experiment, no malformations or lesions were identified in any of the individuals after ingesting the MPs in the diet. Treatment A (control) also showed no morphological abnormalities. Another similar study also reported no increase in deformities in larvae fed with microplastics collected from environmental matrices when compared to larvae fed only fish food [28]. However, none of the studies analyzed the impacts of ingested MPs at the tissue or cellular level. Furthermore, behavioral changes and deleterious effects on reproduction and offspring have also not been investigated.

### 3.2. Ingestion and Excretion of MPs

The number of MPs found in the feces (excreted) 2, 24 and 48 h after feeding, as well as the number of MPs found in the gastrointestinal tract (accumulated) at these same times can be seen in Table 2. In addition, the percentage of MPs still present in the gastrointestinal tract and excreted 48 h after feeding was also calculated (Figure 3). No MPs were found in the feces or in the gastrointestinal tract of individuals of Treatment A (control).

The accumulation and excretion rates 48 h after the last feeding varied around 18.51% ± 3.05 and 77.96% ± 2.62, respectively, for Treatment B; 9.42% ± 1.51 and 89.94% ± 1.74, respectively, for Treatment C; 18.51% ± 3.05 and 77.96% ± 2.62, respectively, for Treatment D; and 18.33% ± 1.47 and 80.23% ± 1.70, respectively, for Treatment E.

Therefore, Treatments B and D, in which the lambari were fed with 0.1% MPs from PE and PET polymers, respectively, showed the lowest excretion rates and, consequently, the highest accumulation rates of MPs ingested 48 h after the last feeding. Studies carried out by Pannetier et al. [28] found similar results, where MP samples were collected and fed, along with the feed, to fish larvae for 30 days, in three different concentrations (0.01, 0.1, and 1.0%). The results found in this study indicate that the ingestion of MPs by the larvae caused death, delayed growth, increased EROD activity, DNA breaks, and changes in swimming behavior, with the diet containing 0.1% MPs having the greatest impact. This evidence corroborates the results found in this study and demonstrates the seriousness of the ingestion of MPs by fish, highlighting the sublethal effects caused in these organisms by the accumulation of realistic concentrations of MPs.

Treatment C, in which the lambaris were fed 1.0% PE polymer MPs, was the treatment that showed the highest excretion rates and, consequently, the lowest accumulation rates of the MPs ingested 48 h after the last feeding. Again, this may be explained by the possibility that, in treatments with higher concentrations of MPs, the fish were able to select particles containing only feed, ingesting fewer feed particles containing MPs [37] and, consequently, accumulating fewer MPs in the gastrointestinal tract. In addition, the short residence time of these particles in the gastrointestinal tract of these animals—evidenced by the high excretion rates over the time analyzed—is a probable explanation for the relatively low proportion of contaminated fish in this treatment, which was the one with the lowest mortality rate when compared to the other treatments. However, studies carried out on marine species indicate that ingesting large quantities of PE particles can cause abdominal distension and abnormal swimming behavior, which in natural conditions would have secondary effects, such as vulnerability to predation [41]. These contradictory results emphasize the need for more research, especially in freshwater ecosystems, since freshwater fish are the animal group with the highest number of records of plastic ingestion [42].

Similarly to Treatment C, Treatment E, in which the lambari were fed 1.0% PET polymer MPs, also showed relatively high excretion rates of the MPs ingested 48 h after the last feeding. However, unlike Treatment C, Treatment E had the highest mortality rate of all the treatments. In agreement, Jakubowska et al. [43] analyzed the effects of chronic exposure (113 days) to MPs from different types of polymers on the early life stages of *Salmo trutta*, and their results showed that exposure to PET polymer MPs affected more significantly the development and induction of endocrine, genotoxic, and cytotoxic responses in sea trout than PE MPs. In addition, PET has a significant potential to leach endocrine disrupters [44,45,46] and, unlike the PE pellets used in this study—virgin and without additives—the PET pellets were post-consumer pellets destined for recycling, which includes the possibility of them having chemical additives on their surfaces. Although this parameter was not assessed in the current study, it may have occurred and contributed to the higher mortality rates observed for this polymer, even though it had the highest excretion rates. However, it is also worth noting that despite PE being reported as a polymer with less toxic potential, according to some studies, it still has considerable sorption affinity for a wide range of organic contaminants available in the environment [47], making its availability in aquatic environments extremely worrying.

## 4. Conclusions

From the results obtained, it can be seen that dietary exposure to MPs, especially PET polymer, was responsible for causing toxicological effects, such as mortality, in the lambari rosa. However, no morphological or weight changes were observed in the individuals analyzed.

The accumulation of MPs in the gastrointestinal tract of the lambari rosa was evident for both polymers in the time interval evaluated, since neither type of plastic was 100% excreted within 48 h of ingestion. We therefore conclude that further studies are needed, considering even longer time intervals, in order to elucidate the gap in the accumulation of MPs by this species. We also encourage other studies to analyze additional types of microplastics, aiming to realistically represent environmental contamination of aquatic ecosystems.

In addition, this study should be combined with research and attempts at environmental mitigation, since the presence of MPs increases the risk of exposure to biota and raises concerns about the availability of these microparticles in aquatic environments, especially freshwater ones. Furthermore, we recommend future research to assess the impacts of MPs on food safety, to improve understanding of the possible toxicological implications and effects on human health.

## Figures and Tables

**Figure 1 animals-13-03363-f001:**
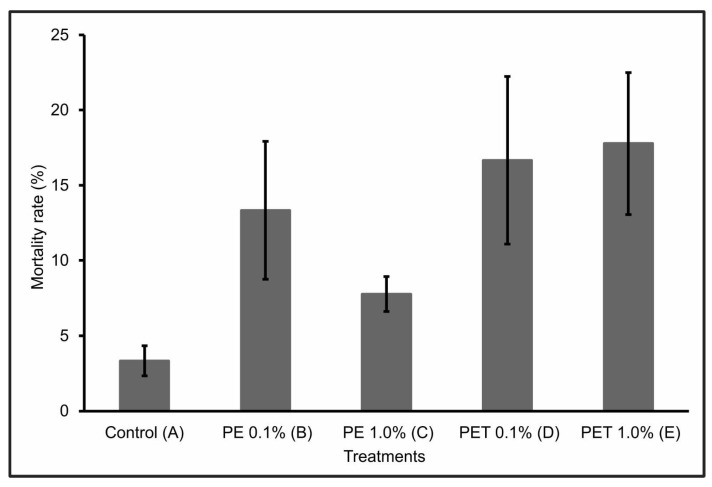
Mortality rate (%) (mean ± standard deviation) of lambari rosa at the end of the 90-day experiment (*n* = 90 individuals). A: only MP-free feed (control); B: feed + 0.1% PE MPs; C: feed + 1.0% PE MPs; D: feed + 0.1% PET MPs; E: feed + 1.0% PET MPs.

**Figure 2 animals-13-03363-f002:**
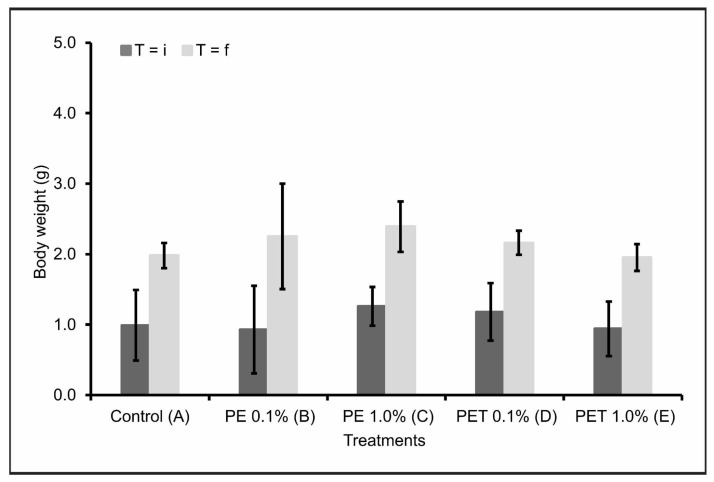
Body weight in grams (mean ± standard deviation) of lambari rosa at the beginning (T = i) and end of the experiment (T = f) (*n* = 90 individuals). A: only MP-free feed (control); B: feed + 0.1% PE MPs; C: feed + 1.0% PE MPs; D: feed + 0.1% PET MPs; E: feed + 1.0% PET MPs.

**Figure 3 animals-13-03363-f003:**
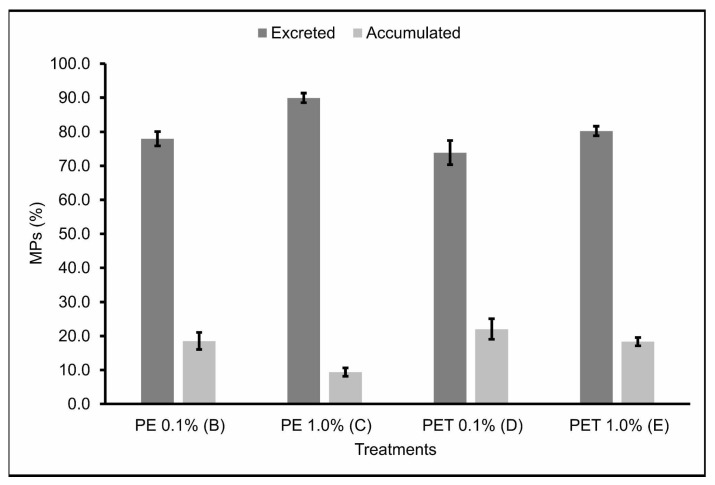
MPs (%) present in the gastrointestinal tract (accumulated) and feces (excreted) (*n* = 45 individuals) 48 h after the last feeding. B: feed + 0.1% PE MPs; C: feed + 1.0% PE MPs; D: feed + 0.1% PET MPs; E: feed + 1.0% PET MPs.

**Table 1 animals-13-03363-t001:** Experimental design.

	Experimental Design	
		Feed
**Treatments**	Control (A)	only tetraMin^®^ extruded feed
	PE 0.1 % (B)	tetraMin® extruded feed and 0.1% PE polymer MPs
	PE 1.0 % (C)	tetraMin® extruded feed and 1.0% PE polymer MPs
	PET 0.1 % (D)	tetraMin® extruded feed and 0.1% PET polymer MPs
	PET 1.0 % (E)	tetraMin® extruded feed and 1.0% PET polymer MPs

**Table 2 animals-13-03363-t002:** MPs found in the gastrointestinal tract (milligrams) and feces (milligrams) at 2, 24 and 48 h after the last feeding (mean ± standard deviation). B: feed + 0.1% PE MPs; C: feed + 1.0% PE MPs; D: feed + 0.1% PET MPs; E: feed + 1.0% PET MPs. Different letters indicate statistical difference by the Tukey test (*p*-value < 0.05).

MPs Found in the Gastrointestinal Tract (mg) and Feces (mg) after Feeding			
Treatments	2 h	24 h	48 h
**MPs in Gastrointestinal Tract (mg) (*n*** ** = 15)**			
PE 0.1% (B)	1.65 ± 0.05 b	0.98 ± 0.06 b	0.33 ± 0.04 b
PE 1.0% (C)	16.93 ± 0.06 a	7.01 ± 0.09 a	1.70 ± 0.22 a
PET 0.1% (D)	1.68 ± 0.02 b	1.05 ± 0.05 b	0.40 ± 0.05 b
PET 1.0% (E)	16.96 ± 0.07 a	7.23 ± 0.03 c	3.30 ± 0.22 c
**MPs in Feces (mg) (*n*** ** = 15)**			
PE 0.1% (B)	0.13 ± 0.05 b	0.66 ± 0.02 b	0.62 ± 0.02 b
PE 1.0% (C)	1.20 ± 0.29 a	9.98 ± 0.06 a	5.29 ± 0.15 a
PET 0.1% (D)	0.09 ± 0.01 b	0.61 ± 0.05 b	0.62 ± 0.03 b
PET 1.0% (E)	0.98 ± 0.04 a	9.70 ± 0.08 a	3.77 ± 0.29 c

## Data Availability

The data presented in this study are available within the article.

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
