# Peer review of "Effects Caused by the Ingestion of Microplastics: First Evidence in the Lambari Rosa (Astyanax altiparanae)"

_animals, 2023, doi:10.3390/ani13213363_

Round 1

Reviewer 1 Report

Introduction:

When introducing "Microplastics (MPs)" for the first time, it's a good practice to provide the full term in parentheses. For instance, "Microplastics (MPs) can be defined..."

Ensure that each sentence conveys a single, clear point. Some sentences are quite long and could benefit from simplification. For instance, "The textile industry utilizes both natural fibers, such as cotton and wool, and synthetic fibers, such as polyester (polyethylene terephthalate (PET)), nylon (polyamide (PA)), and elastane (85% polyurethane (PU))." could be broken into shorter sentences for clarity.

Line 38: Any reference for this statement?

Line 39–40: The clarity of this statement is missing; why is it a great concern for the scientific community?

Line 63: Include the reference at the end of the sentence.

Please define terms that might be unfamiliar to a broader audience, like "lambari rosa," to enhance reader understanding.

Mention the scientific name of "Lambari Rosa." I know it is mentioned in abstract, it will be better for the readers if it also mentioned first time in the body of manuscript.

In the last paragraph, clarify what the unique contribution of your study is. What makes it different from existing research on similar topics?

Materials and Methods

The experimental treatments are confusing; it will be easy to see if the experimental design is in the table format.

It will be better to use only one software for the data analysis of the study, "R" software, for better image quality of the graphs.

The data was analyzed using ANOVA (one-way). What is the reason for utilizing it, and what are the variables that you are choosing for the analysis?

Results and discussion

To be consistent, please include the details of the data "(17.77% ± 1.48)" in this format.

The quality of the graphs is too low and needs improvement.

In the figure legends, please include the sample size used for the data analysis, for example (n = 5–8).

Figure 3 is too busy and hard to understand. Please improve the quality of the graph.

Sections 3.1.3 and 3.1.2: Why do both sections not have any discussion points?

Section 3.2 is very hard to follow because of the results and discussion. Please make it clearer and more concise.

Conclusion

Are there any limitations for this study?

How can these toxicological results from this study be compared to those of humans?

The quality of the writing is satisfactory; however, there are instances where the sentences are excessively lengthy and challenging to comprehend.

Reviewer 2 Report

Linhas 47 a 48: Existem parênteses duplos na escrita "))".

Linha 78: Escreva a espécie do lambari rosa. Apesar de citar o gênero na linha 76, na linha 78 o lambari rosa é citado sem indicar sua espécie (Astyanax altiparanae).

Linhas 86 a 92: Deixar mais claro qual foi o objetivo da pesquisa.

Linha 95: Qual o peso médio dos peixes?

Linha 99: Após essa aclimatação inicial, os peixes continuaram se adaptando às novas condições ambientais por quantos dias antes de receberem a dieta experimental?

Linha 100: como foi feita a distribuição nos aquários? As informações da linha 112 devem ser inseridas aqui.

Linhas 103 e 104: Esta informação não se refere à metodologia. Sugiro adicioná-lo à introdução, ao lado do parágrafo sobre Lambari (linhas 76 a 82).

Linhas 106 a 110: Sugiro retirar os nomes dos tratamentos (A, B, C, D, E) ao longo do texto do manuscrito e usar apenas os nomes dos compostos adicionados à dieta. Exemplo: “Foram realizados cinco tratamentos experimentais, constituídos por dietas contendo ração tetraMin® extrusada com adição dos polímeros microplásticos PE (polímero polietileno) e PET (tereftalato de polietileno). Os tratamentos foram: dieta sem adição de polímeros microplásticos (controle), e dietas com 0,1% de PE; 1% de PE, 0,1% de PET e 1% de PET."

Linha 222: Na Tabela 1 sugiro apresentar os valores médios (Ex: média dos valores descritos como “B” (B1+B2+B3/3)).

Linhas 287 a 290: Esta informação deve constar apenas em “conclusões”. Contudo, a informação é repetida nas conclusões, nas linhas 296 e 297. Portanto, estas linhas 287 a 290 devem ser removidas.

Round 2

Reviewer 1 Report

No comments.